# Detection of Intestinal Parasites in Stray Dogs from a Farming and Cattle Region of Northwestern Mexico

**DOI:** 10.3390/pathogens9070516

**Published:** 2020-06-28

**Authors:** Enrique Trasviña-Muñoz, Gilberto López-Valencia, Francisco Javier Monge-Navarro, José Carlomán Herrera-Ramírez, Paulina Haro, Sergio Daniel Gómez-Gómez, Julio Alfonso Mercado-Rodríguez, Cesar Augusto Flores-Dueñas, Sergio Arturo Cueto-Gonzalez, Mariel Burquez-Escobedo

**Affiliations:** 1Institute of Research in Veterinary Sciences, Autonomous University of Baja California, Carretera, Mexicali-San Felipe Km 3.5, Laguna Campestre, Mexicali 21386, Mexico; etrasvina@uabc.edu.mx (E.T.-M.); fmonge@uabc.edu.mx (F.J.M.-N.); jherrera20@uabc.edu.mx (J.C.H.-R.); gomez.sergio@uabc.edu.mx (S.D.G.-G.); juliomr@uabc.edu.mx (J.A.M.-R.); augusto.flores@uabc.edu.mx (C.A.F.-D.); sergiocueto@uabc.edu.mx (S.A.C.-G.); mariel.burquez@uabc.edu.mx (M.B.-E.); 2Dr. Hideyo Noguchi Regional Research Center, Autonomous University of Yucatan, Av. Itzáes 490, Centro, 97000 Mérida, Mexico; paulina.haro@correo.uady.mx

**Keywords:** intestinal parasites, zoonoses, public health, Mexico, stray dogs

## Abstract

Stray dogs are one of the main reservoirs of intestinal parasitic infections and some have zoonotic potential. An epidemiological survey was carried out between September 2017 and May 2018 in Mexicali Valley, this area sacrifices around 92,470 head of cattle monthly, which represents 27% of the national slaughter and has 71,307 hectares for crops. In this period the Municipal Animal Control Center during their routine visits to the Mexicali Valley captured 103 dogs. All the dogs were evaluated using copromicroscopic techniques to detect intestinal parasites. The general frequency of parasitic infections was 28.15% (29/103), the most frequent parasite being *Dipylidium caninum* 16.50% (17/103), followed by *Taenia* spp. 6.79% (7/103), *Taenia hydatigena* 2.91% (3/103), *Taenia serialis* 0.97% (1/103), *Taenia pisiformis* (0.97%)*,*
*Toxocara canis* 3.88% (4/103), *Toxascaris leonina* 1.94% (2/103), and *Cystoisospora* spp. 1.94% (2/103). No significant statistical associations were found between parasitic infections and the studied variables (sex, age, and size) however; there was a significant statistical association with the capture area. Most of the parasites found in this survey have potential to affect the human population and animal production.

## 1. Introduction

Human activities cause alterations in the ecosystem that may result in negative consequences for the health of humans and many animal species [1]. Changes caused by urbanization, farming, and livestock activities along with the increase in population density, the lack of programs for the surveillance of many high-impact diseases and the absence of preventive medicine strategies for human and animal populations promote favorable ecological conditions for the development of parasitic diseases of public health importance in those populations [2]. Stray dogs represent one of the most important animal species that serve as a host and reservoir for parasites of public health importance, dogs excrete larvae, oocysts and eggs of intestinal and pulmonary parasites [3]. Contamination of the environment through animal feces increases the risk for the transmission of parasitic infections to humans and other animal species. When contaminated plant or animal food products are consumed cause parasitic infections with potentially severe complications [4]. In the northwest region of Mexico, the infections produced by intestinal parasites have been previously reported in the stray dog population from the city of Mexicali and are considered as an important public health problem [5]. The aim of the survey was to establish the frequency and geographical distribution of intestinal parasitic infections in stray dogs from a farming and cattle region of the Mexicali Valley and to demonstrate the risk of intestinal parasites to livestock and agriculture activities.

## 2. Materials and Methods

### 2.1. Animal Ethics and Welfare

All animal handling procedures were conducted following national code NOM-033-SAG/ZOO-2014. All procedures were also reviewed and approved by the Institutional Committee for Animal Welfare of the Academic Group for Diagnosis of Infectious Diseases of the Institute for Research in Veterinary Sciences (IICV) of the Autonomous University of Baja California (UABC).

### 2.2. Sample Collection and Parasitological Procedures

A cross-sectional epidemiological survey was conducted in the Mexicali Valley, located at 32°37′40′′ N and 115°27′16′′ W in the Northwest corner of Mexico and south of the borderline with the state of California in the USA. Mexicali Valley has an expansion of approximate 370,900 hectares and a desert climate where summer temperatures can reach 45 to 50 °C [6]. As for the agricultural production, is an area that sacrifices 92,470 head of cattle monthly, which represents 27% of the national slaughter and has 71,307 hectares for crops [7,8]. The present survey was performed between September 2017 and May 2018 in dogs impounded by the Municipal Animal Control Center (CEMCA) of Mexicali. Impounded dogs are held at CEMCA for 3 to 7 days before they are sold, adopted or euthanized. The euthanasia protocol consists of an initial dose for deep sedation followed by an anesthesia overdose. A total of 103 stray dogs were captured and euthanized by personnel of CEMCA during their rounds to the Mexicali’s agricultural valley were included in this survey. Individual information of each impounded dog was collected including sex (male or female), dental age established as younger or older than one year, size (small, medium, or large), and zone of capture within the Mexicali Valley (eastern, central or western) and the presence of ectoparasites associated with intestinal parasites. Following euthanasia, dogs were dissected directly at CEMCA premises to collect the small intestine and cecum. Stool samples were collected directly from the large intestine and rectum. Collected materials were placed in tagged plastic bags, kept at 4 ºC in blue ice, and sent to the laboratory for detection and identification of intestinal parasites. For parasitology procedures, a longitudinal incision was performed on the small intestine and cecum and the contents initially analyzed to direct visualization of adult parasites, intestinal content was then processed to visualize eggs and oocysts using the zinc sulfate flotation technique (specific gravity 1.18) and Lugol’s iodine solution for the identification of protozoan cysts and coccidial oocysts [9]. Egg and oocyst counts were performed using the McMaster technique (with zinc sulfate solution and 1.18 specific gravity solution as well) and the identification of helminth species, eggs, and oocysts was based on the morphological characteristics described by Zajac and Conboy [10].

### 2.3. Molecular Identification of Taenia Species

To determine the specific *Taenia* species, DNA was extracted from proglottids using the DNeasy Blood & Tissue Kit (QIAGEN, Valencia, CA, USA), and endpoint polymerase chain reaction (PCR) was performed targeting a fragment of 446 base pairs (bp) of the mitochondrial cytochrome c oxidase subunit 1 (mt-CO1) gene following the procedure described by Utuk and Piskin [11]. The PCR products were separated on agarose gel (1.5%) and stained with ethidium bromide and visualized on an UV transilluminator. Also, the PCR product was purified, sequenced and compared with sequences published in GenBank, using the BLAST tool (https://blast.ncbi.nlm.nih.gov/Blast.cgi).

### 2.4. Statistical Analysis

The frequencies of the overall cases of parasitic intestinal infections from each capture zone, specific parasite, single and multiple-infected samples, and counting of oocysts or eggs per gram (EPG) were calculated. Chi-square (χ²) was calculated to establish associations between parasitic infections and the analyzed variables, and odds ratios (ORs) were also calculated with 95 % confidence intervals. All statistical calculations were performed using the software Statistix 9^®^ (Analytical Software, Tallahassee, FL, USA).

## 3. Results

The results of the parasitological analysis showed that 29/103 (28.15%) samples of feces and intestines were positive for intestinal parasites, being *Dipylidium caninum* the most frequent parasite found in 17/103 (16.50%) samples, followed by *Taenia* spp. 7/103 (6.79%), *Toxocara canis* 4/103 (3.88%), *Toxascaris leonina* 2/103 (1.94%), and *Cystoisospora* spp. 2/103 (1.94%). The results also demonstrate three cases of co-infection with *D. caninum* and *Taenia* spp. among all the positive cases (Table 1). The PCR results for *Taenia* spp. demonstrate amplification of the target mt-CO1 gene in 5/7 samples. Those five PCR products were sequenced and compared with published sequences, showing that 3/5 matched the sequences of *Taenia hydatigena* (100% identity) under GenBank accession numbers: MK851045.1, KY012314.1, and MN175597.1; 1/5 matched the sequence of *Taenia serialis* (99% identity) according to GenBank accession number: MH350844.1, and 1/5 matched the sequence of *Taenia pisiformis* (92% identity) under GenBank accession number: GU569096.1 (Table 2).

No significant statistical associations were found between parasitic infections and the studied variables such as sex, age, and size, but it is important to note that not a single dog infected with *Taenia* was of small size or younger than 1 year (Table 3, Table 4 and Table 5), however; there was a significant statistical association with the capture area (Table 6). In the western zone of the Mexicali Valley, more cases of dipylidiosis were detected than in the rest of the zones (*p* < 0.01) and cases of toxocarosis were only found in the east zone (*p* < 0.05).

## 4. Discussion

Our survey demonstrates the presence of intestinal parasitic infection with zoonotic implications in almost three out of every ten stray dogs sampled from the Mexicali Valley. The problem of parasitic intestinal infections transmitted by stray dogs continues to have an important public health impact not only in northwestern Mexico but also as a sanitary problem reported in other regions of our country, where the presence of parasitic intestinal infections are more widespread in the stray dog’s populations, such as the study reported by Cortez-Aguirre [12] in the city of Campeche, in the southeast region of Mexico seeking for eggs of intestinal parasites in public parks where stray dogs roam freely and were observed defecating. The results of that study showed that all (100%) public parks studied were found to be contaminated with dog feces harboring eggs of intestinal parasites, mainly from *Ancylostoma caninum*, *T. canis*, and *D. caninum*. In a similar study conducted in household dogs that are allowed to roam freely in the streets of a rural community of the Yucatan peninsula, also in the southeast region of Mexico, and performed to establish the prevalence of canine intestinal parasites with the potential zoonotic transmission, found that the dogs of that locality tested positive for either *A. caninum*, *Trichuris vulpis*, *T. canis*, and *D. caninum* for an overall prevalence of 80% with mixed infections caused by two or more parasite species observed in over 50% of the samples analyzed [13]. The transmission of intestinal parasitic infections has also been reported in other Latin American countries where the presence of intestinal parasites of zoonotic importance has been demonstrated in stray dogs from urban areas. A study from Brazil seeking the presence of helminths in stray dogs living in urban areas showed that more than 90% of those animals were heavily parasitized, being *A. caninum* and *D. caninum* the most prevalent parasites found, with less than 10% of examined dogs were found negative for parasites of public health importance [14]. In a similar study conducted in Argentina seeking to determine the prevalence of intestinal parasites in the feces of stray dogs found an overall prevalence of 89.0%, intestinal parasites in the samples analyzed, being *A. caninum*, *T. canis* and *D. caninum* the most prevalent parasites detected with multiple parasite infections detected in 80% of the animals tested, indicating that the feces from stray dogs are an important source of several zoonotic parasitic infections of that Latin American region [15]. It is evident that climatic conditions of temperature and soil type of the southeast region of Mexico and countries in the rest of Latin America are more suitable to support a diverse range of parasitic infections of zoonotic importance, reflected in higher prevalence rates of parasitic infections [16]. We believe that *D. caninum* and *Taenia* spp. were the most frequent parasites since, despite the extreme climatic conditions of Mexicali Valley, these parasites fulfill most of their biological cycle within the intermediate and definitive host [17]. *T. canis* was also one of the most frequent parasites detected, this nematode has a thick shell that enables egg survival in the external environment for many years and confers them resistance to the harsh environmental conditions in soil [18].

Regarding the distribution of intestinal parasitic infections, the results showed that *D. caninum* was the most frequent parasite detected in the majority of the western zone (*p* < 0.01) of the Mexicali Valley. This parasitic infection affects dogs, cats, and occasionally humans, and is acquired through the ingestion of the intermediate host, fleas of the species *Ctenocephalides canis*, *Ctenocephalides felis*, *Pulex irritans,* or the dog louse *Trichodectes canis* [19]. However, in the present work, no dog had any ectoparasites that served as an intermediate host for *D. caninum*, this could be explained by the fact that applied by personnel from the Municipal Control Animal Center, there is a permanent fumigation program based on the use of deltamethrin around the areas where the dogs are confined, further studies will be required to identify the intermediary host for this parasite in the region.

Infections produced by different species of *Taenia* were also detected in this work. This parasitic infection was previously reported in this region but those studies lack the species identification of the specimens detected. In our work, three species of *Taenia* were identified using molecular techniques: *T. pisiformis*, *T. serialis*, and *T. hydatigena*. The first two were found in the east area of the Mexicali valley where grains and forages are the dominant agricultural crops, with an abundance of lagomorphs such as rabbits and other rodents associated with the biological cycle of *T. serialis* and *T. pisiformis* [20,21]. Dog gets infected while hunting and consumes the intermediate host, acquiring the metacestodes and completing the biological cycle of these parasites [22]. It is important to mention that dog infection with *T. serials* represents a public health problem, accidental human consumption of eggs, can develop the juvenile stage (coenuri) in various tissues such as eyes, brain but mainly subcutaneous tissue [23].

Also, three cases of *T. hydatigena* were detected in the central area of the Mexicali Valley. The definitive host for this parasite are dogs that become infected by the ingestion of tissues of the intermediate hosts that contain the metacestode of *T. hydatigena*. Cattle, sheep, goats, pigs, and wild cervids act as intermediary host of *T. hydatigena* where the larva stage, the cysticercus, develops and complete its biological cycle by passing through the wall of the intestine and finds its path to the liver, the peritoneal cavity and other tissues [24]. The presence of *T. hydatigena* in the central area of the Mexicali valley might be explained by the fact that the feedlot operations and dairy farms are located. Monetary losses due to parasite carcass confiscation have been reported to be, up to 370,000 US dollars per year [25].

All cases of infection with *T. canis* were detected in the eastern area of the Mexicali Valley (*p* < 0.05), with an average of 1075 eggs per gram of feces. *T. canis* represents an important threat to human health since the eggs are eliminated with the feces and can directly produce infection in both dogs and humans. In humans, *Toxocara* can cause two syndromes: visceral larva migrans and ocular larva migrans, both of which with important health consequences [26]. The east part of the Mexicali Valley is also a region for horticultural activities where most of the recollection of crops is done by hand, with an increased risk of direct contact with infected feces and/or direct contamination of vegetables. In Mexicali Valley there are no reports of toxocarosis in humans, in fact in the entire country of Mexico there are no reports of this disease by the government health sector, there is a lack of specific diagnostic programs in our country and we believe that greater importance should be given to this zoonotic pathogen as not only is found in this survey, in the city of Mexicali a survey carried out in parks found that 54% of them were contaminated with *T. canis*, being children one of the main population at risk because they have more contact with soil [27].

## 5. Conclusions

The data obtained from this work, is relevant to public health, it was shown that most of the parasites detected are zoonotic, with *D. caninum* being the most prevalent parasite, in addition to other parasites which can affect the health of the population such as *T. canis.* Also *T. hydatigena* was found in this work, demonstrating indirectly that livestock animals in this area are affected as they need to carry the juvenile parasite in order to complete the biological cycle. It is necessary to carry out an epidemiological study in this area where the intermediary hosts of cestodes are included in order to have a better understanding of the distribution and risk factors of these parasites.

## Figures and Tables

**Table 1 pathogens-09-00516-t001:** Frequency of parasitic infections and the average number of eggs per gram of feces (EPG) diagnosed using copromicroscopic techniques in stool samples from the intestine of stray from the Mexicali Valley.

Detected Parasites	Positive/Analyzed	Average of Oocysts or Eggs per Gram of Feces	Frequency (%)
*Dipylidium caninum*	14/103	-	13.59
*Taenia* spp.	4/103	-	3.88
*Taenia hydatigena*	3/103	-	2.91
*Taenia serialis*	1/103		0.97
*Taenia pisiformis*	1/103		0.97
*Toxocara canis*	4/103	1075	3.88
*Toxascaris leonina*	2/103	300	1.94
*Cystoisospora* spp.	2/103	600	1.94
***Coinfection***			
*Dipylidium + Taenia* spp.	3/103		2.91
Total	29/103		28.15

**Table 2 pathogens-09-00516-t002:** Pairwise alignment of *Taenia pisiformis* and *Taenia serialis* collected from the intestines of stray dogs from the Mexicali Valley.

Sample	Start	Sequence Comparison: Reference/Detected	End
GU569096.1 *	7623	TATTGTTTGCAATGTTTTCTATAGTTTGTTTAGGTAGAAGTGTATGAGGTCATCATATGT	7682
Sequence **	1	TATT**A**TTTGCAATGTTTTCTAT**T**GTTTGTTTAGGTAGAAGTGTATGAGG**C**CATCATATGT	60
GU569096.1	7683	TTACTGTTGGATTAGATGTAAAGACCGCTGTGTTTTTTAGTTCAGTAACAATGATAATTG	7742
Sequence	61	TTACTGTTGG**G**TTAGATGTAAAGAC**T**GC**C**GT**A**TTTTTTAGTTC**G**GTAACAATGATAATTG	120
GU569096.1	7743	GAGTACCTACTGGAATTAAGGTCTTTACATGACTTTATATGCTTTTAAATTCTCGTGTCA	7802
Sequence	121	GAGTACC**A**AC**A**GGAATTAAGGT**T**TTTACATG**G**CTTTA**C**ATGCTTTTAAATTCTCGTGT**T**A	180
GU569096.1	7803	AAAAGAGTGATCCTGTGTTGTGGTGAATAATTTCTTTTATAGTCTTATTTACTTTTGGAG	7862
Sequence	181	AAAAGAGTGATCCT**A**T**A**TTGTG**A**TGAATAATTTCTTTTATA**A**T**T**TT**G**TTTACTTTTGG**T**G	240
GU569096.1	7863	GTGTAACTGGTATAGTATTATCTGCTTGTGTTTTAGATAAAGTTTTACATGATACTTGAT	7922
Sequence	241	GTGTAAC**A**GGTATAGT**T**TTATCTGC**C**TGTGT**G**TTAGATAAAGTTTTACATGATACTTGAT	300
GU569096.1	7923	TTGTTGTAGCGCATTTTCATTATG	7946
Sequence	301	TTGTTGT**G**GC**T**CATTTTCATTATG	324
* *Taenia pisiformis* mitochondrion, complete genome sequence, reference ID: GU569096.1 ** Identities: 297/324 (92%), Gaps: 0/324 (0%), Strand: Plus/Plus			
MH350844.1 *	85	TTGTTATTTGCTATGCTCTCAATAGTGTGTTTAGGAAGGAGTGTATGGGGTCATCATATG	144
Sequence **	1	TTGTTATTTGCTATGCTCTCAATAGTGTGTTTAGGAAGGAGTGTATGGGGTCATCATATG	60
MH350844.1	145	TTTACAGTTGGGTTAGATGTTAAGACTGCTGTATTTTTTAGCTCAGTTACTATGATAATA	204
Sequence	61	TTTACAGTTGGGTTAGAT**A**TTAAGACTGCTGTATTTTTTAGCTCAGTTACTATGATAATA	120
MH350844.1	205	GGAGTACCAACAGGAATAAAGGTTTTTACTTG	236
Sequence	121	GGAGTACCAACAGGAATAAAGGTTTTTACTTG	152
* *Taenia serialis* cytochrome c oxidase subunit I gene, reference ID: MH350844.1 ** Identities: 151/152 (99%), Gaps: 0/152 (0%), Strand: Plus/Plus			

Nucleotides in bold and underlined are different from the compared sequences.

**Table 3 pathogens-09-00516-t003:** Frequency of the diagnosis of parasitic infections according to the sex of stray dogs from the Mexicali valley.

Total (n = 103)	Male % (n = 38)	Female % (n = 65)	*p*
*Dipylidium caninum*	21	13.8	0.34
*Taenia* spp.	7.8	6.1	0.73
*Taenia hydatigena*	2.6	3.0	0.89
*Taenia serialis*	2.6	0	0.18
*Taenia pisiformis*	0	1.5	0.44
*Toxocara canis*	0	6.1	0.11
*Toxascaris leonina*	0	3.0	0.27
*Cystoisospora*	0	3.0	0.27
Overall prevalence	26.3	29.2	0.75

Comparison of the general and specific frequency by sex. Results of the calculation of χ².

**Table 4 pathogens-09-00516-t004:** Frequency of the diagnosis of parasitic infections according to the size of stray dogs from the Mexicali valley.

Total (n = 103)	Small % (n = 22)	Medium % (n = 58)	Large % (n = 23)	*p*
*Dipylidium caninum*	13.6	12	30.4	0.12
*Taenia* spp.	0	6.89	13	0.22
*Taenia hydatigena*	0	3.4	4.3	0.64
*Taenia serialis*	0	1.7	0	0.67
*Taenia pisiformis*	0	0	4.3	0.17
*Toxocara canis*	9	3.4	0	0.27
*Toxascaris leonina*	0	1.7	4.3	0.56
*Cystoisospora*	4.5	1.7	0	0.53
Overall prevalence	27.2	25.8	34.7	0.71

Comparison of the general and specific frequency by size. Results of the calculation of χ².

**Table 5 pathogens-09-00516-t005:** Frequency of the diagnosis of parasitic infections according to the age of stray dogs from the Mexicali valley.

Total (n = 103)	Younger than 1 Year % (n = 13)	Older than 1 Year % (n = 90)	*p*
*Dipylidium caninum*	7.6	17.7	0.35
*Taenia* spp.	0	7.7	0.29
*Taenia hydatigena*	0	3.3	0.50
*Taenia serialis*	0	1.1	0.70
*Taenia pisiformis*	0	1.1	0.70
*Toxocara canis*	7.6	3.3	0.44
*Toxascaris leonina*	0	2.2	0.58
*Cystoisospora*	7.6	1.1	0.10
Overall prevalence	23	28.8	0.66

Comparison of the general and specific frequency by dental age. Results of the calculation of χ².

**Table 6 pathogens-09-00516-t006:** Frequency of the diagnosis of parasitic infections according to the capture zone of stray dogs from the Mexicali valley.

Total (n = 103)	West Zone % (n = 7)	Central Zone % (n = 59)	East Zone % (n = 37)	*p*
*Dipylidium caninum*	42.8	22	2.7	0.006 **
*Taenia* spp.	0	6.7	8.1	0.73
*Taenia hydatigena*	0	5	0	0.31
*Taenia serialis*	0	0	2.7	0.40
*Taenia pisiformis*	0	0	2.7	0.40
*Toxocara canis*	0	0	10.8	0.02 *
*Toxascaris leonina*	0	1.6	2.7	0.87
*Cystoisospora*	0	0	5.4	0.16
Overall prevalence	42.8	27.1	27	0.66

Comparison of the general and specific frequency by catch area. Results of the calculation of χ². * *p* < 0.05, ** *p* < 0.01.

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
