# Peer review of "Detection of Intestinal Parasites in Stray Dogs from a Farming and Cattle Region of Northwestern Mexico"

_pathogens, 2020, doi:10.3390/pathogens9070516_

Round 1

Reviewer 1 Report

The article studies the presence of Intestinal Parasites in Stray Dogs from a Farming and Cattle Region of Northwestern Mexico. The results of the study are very interesting because they provide actual data about pathogens that are of great importance human and animal health. 

The structure of the manuscript is generally well defined although sometimes the information is repeated and/or it is not well defined. Ie. In the abstract there is information that does not appear in the text or some information included in the material and methods section is not provided later in the text.

Regarding the results, the organization could be improved. In my opinion, when describing of discussing several types of parasites they should be considered in a separated way.  It is not good consider in the same level protozoan as nematodes or cestodes.

The results showed the infection by several parasite species but not the diseases produced by them. It should be more correct to refer a Dipylidium infection / egg output…. Many information is repeated in the text and tables.

In the discussion author compare their results and those of other studies but generally do not try to explain the differences.

Some suggestions:

Abstract

Please, indicates the Taenia species individually

The names of the species in cursive

The information about the study area is not included later into the material and methods section

Key words

Teniasis is not enough to reflect the results of the study. Better:  intestinal parasites

Introduction

  • Line 36       “of intestinal parasites, larvae, and eggs “   suggest…..  larvae, and eggs of intestinal  and pulmonary parasites
  • Line 37 delete ecological  (the environment will be contaminated anyway)
  • It would be welcome a final sentence with the aim of the study

Material and methods

  • Line 170 Institute
  • Line 193 Taenia in cursive
  • Line 195
    • what sample was used to determine the Taenia species?
    • How identify Taenia spp when specie identification was not possible?

Results

  • Line 56
    • it would be more correct to add the author that described each specie the first time that it is named
    • caninum 17/103 (16.50%)  but in the table 1  14/103 (13.59%)
  • in all the text the second time the initial of the genera is enough: A.caninum….
  • Lines 18 and 57, table 1             Taenia spp included several species so in the same list species and genus should not be mixed. If identification was done until species level, all the species should be specifically noted
  • Table 1        Cystoisospora eliminates oocysts, no eggs
  • Table 2 if the secuencies are published in the references or in the Genbank this table is not necessary. Or it should be justified
  • Table 3 please, consider to separate species by cestodes, nematodes and protozoan
  • The data about sex, age and size would be interesting to be noted. Please consider to include them in tables 1 and/or 3 despite of no differences were found

Discussion

  • Lines 79-84 this information is practically the same that in the introduction.                Delete here or merge both in the introduction section
  • Line 99 two or more types of parasites? Two or more parasite species?
  • Line 103 presence of
  • Line 105 of examined dogs were found
  • Line 112 delete double spaces in Latin America
  • Lines 111-115 two long sentence
  • Lines 119-120 “is acquired through the ingestion of the intermediate host, fleas of the species”  …….   Is the same that  “all of them involved in the transmission cycle of Dipylidium caninum
  • Line 120 “However, in the present work, we were not able to detect any fleas or louse in the dogs detected to be parasitized”   in the methodology it is not stated that ectoparasites were investigated
  • Line 132 Taenia  in cursive
  • Line 144 Stray dogs
  • Line 148      canis better than Toxocara
  • Line 151 “in the human host”  better: in humans
  • Line 157 “since not all parasites can coexist in the same host due to the competition for nutrients from the intestinal tract”  this assessment is not completely true. In fact, the prevalence of mixed infections is usually higher than monospecific infections

Conclusion

  • It is not necessary to repeat the data of prevalence

References

  • The year in some references is not bold

Reviewer 2 Report

Dear Authors, your manuscript is clear and well written. I made several comments in the pdf.

  • Be careful of the spelling of parasite species. After first use, you can reduce and only write T.canis, A.caninum, D.caninum….but do the same for all, why some entire and some short. spp. is not in italic.
  • I did not understand first that you did necropsy and not only coproscopy. That's why you find so many tapeworms. But I am very surprised by the absence of Ancylostoma and Trichuris, no one, when all other surveys found them. Could you explain.
  • I also do not understand the absence of ectoparasites, were they really research
  • It is not a study, it is a survey: epidemiological survey
  • You go too much on cattle infection, production...All has to be removed, no link. All your dog parasites are source for companion dogs, and finally only Toxocara is of importance for humans, Dipylidium a little bit (it is exceptionnal to ingest a flea). NO risk for Taenia that you found. There is no impact for meat, no zoonotic transmission. Eggs of Toxocara are in the environment, on vegetables, no impact fr food (meat) production. Only economical impact of Taenia hydatigena.
  • Diseases in dogs are Dipylidioisis, Toxocarosis, ascaridosis….osis, "iasis" doesn't exist in a scientific nomenclature.

I think a moderate revision is needed, especially by removing all aspects on cattle and food, which have no real link with the parasites you found. Insist more on the risk for companion dogs, and kids in villages around.

Reviewer 3 Report

The paper  by Trasviña-Muñoz et al. "Detection of Intestinal Parasites in Stray Dogs from a Farming and Cattle Region of Northwestern Mexico" reports the results of an epidemiological study on stray dogs. The English language is to be reviewed and many informations are lacking.

Specific comments:

The title is very promising, but you didn't analyze a real correlation with parsitological problem in cattle. Do you know the prevalence of cysticerosis in the intermediate hosts in this region? 

Moreover you highlighted the potential risk for humans of different parasites. Do you know the data of these parasitosis in humans?

In materials and methods it is no clear how many dogs were euthanised? Did you find a good corrispondence between adult parasites and eggs? What was the density of the flotation solution that you used?What flotation solution did you use from McMaster?Did you use a flotation solution + Lugol for the identification of eggs, cysts and oocysts and to count the McMaster?why didn't you use directly the McMaster? it si not clear.

Tab.2 is very confusing. Please report in the text the prevalence of identity with sequence in GenBank and the nucletide that are different. Moreover explain because you obtained only 5 sequence of 7 samples.

After the first time that you name a parasite you cane write in the abbreviated form, e.g. Tania serialis then T. serialis

Change "coproparasitoscopic" to "copromicroscopic"

Round 2

Reviewer 3 Report

Dear Authors,

compliments for the improvement of your paper.

Some few changes are required:

Materials and methods

  • How did you extract the DNA was from proglottids? Did you use a kit?
  • It is not correct add this sequence (GenBank accession number: JN827307) in materials and methods
  • Lines 96-98: Please add more information on your PCR. The gel that you used was 1% or 2% or another percentage of agarose?
  • Change “The PCR product was DNA purified, sequenced, and compared with published sequences using the BLAST tool (https://blast.ncbi.nlm.nih.gov/Blast.cgi).” to “The PCR product was purified, sequenced and compared with sequences published in GenBank, using the BLAST tool (https://blast.ncbi.nlm.nih.gov/Blast.cgi)”

In table 2: There aren’t the GenBank reference number of the compared sequences, please add them.

In table 2: Remove the sequences with 100% identity.

In table 2: There aren’t the reference nucleotides for the sequences, please add the start and end nucleotide of the sequence compared.
